# Bayesian Nonlinear Support Vector Machines and Discriminative Factor Modeling

**Ricardo Henao, Xin Yuan and Lawrence Carin**
Department of Electrical and Computer Engineering
Duke University, Durham, NC 27708
{r.henao,xin.yuan,lcarin}@duke.edu

## Abstract

A new Bayesian formulation is developed for nonlinear support vector machines (SVMs), based on a Gaussian process and with the SVM hinge loss expressed as a scaled mixture of normals. We then integrate the Bayesian SVM into a factor model, in which feature learning and nonlinear classifier design are performed jointly; almost all previous work on such discriminative feature learning has assumed a linear classifier. Inference is performed with expectation conditional maximization (ECM) and Markov Chain Monte Carlo (MCMC). An extensive set of experiments demonstrate the utility of using a nonlinear Bayesian SVM within discriminative feature learning and factor modeling, from the standpoints of accuracy and interpretability.

## 1 Introduction

There has been significant interest recently in developing discriminative feature-learning models, in which the labels are utilized within a max-margin classifier. For example, such models have been employed in the context of topic modeling [1], where features are the proportion of topics associated with a given document. Such topic models may be viewed as a stochastic matrix factorization of a matrix of counts. The max-margin idea has also been extended to factorization of more general matrices, in the context of collaborative prediction [2, 3]. These studies have demonstrated that the use of the max-margin idea, which is closely related to support vector machines (SVMs) [4], often yields better results than designing discriminative feature-learning models via a probit or logit link. This is particularly true for high-dimensional data (e.g., a corpus characterized by a large dictionary of words), as in that case the features extracted from the high-dimensional data may significantly outweigh the importance of the small number of labels in the likelihood. Margin-based classifiers appear to be attractive in mitigating this challenge [1].

Joint matrix factorization, feature learning and classifier design are well aligned with hierarchical models. The Bayesian formalism is well suited to such models, and much of the aforementioned research has been constituted in a Bayesian setting. An important aspect of this prior work utilizes the recent recognition that the SVM loss function may be expressed as a location-scale mixture of normals [5]. This is attractive for joint feature learning and classifier design, which is leveraged in this paper. However, the Bayesian SVM setup developed in [5] assumed a linear classifier decision function, which is limiting for sophisticated data, for which a nonlinear classifier is more effective.

The first contribution of this paper concerns the extension of the work in [5] for consideration of a kernel-based, *nonlinear* SVM, and to place this within a Bayesian scaled-mixture-of-normals construction, via a Gaussian process (GP) prior. The second contribution is a generalized formulation of this mixture model, for both the linear and nonlinear SVM, which is important within the context of Markov Chain Monte Carlo (MCMC) inference, yielding improved mixing. This new construction generalizes the form of the SVM loss function.

The manner we employ a GP in this paper is distinct from previous work [6, 7, 8], in that we explicitly impose a max-margin-based SVM cost function. In the previous GP-based classifier design, all data contributed to the learned classification function, while here a relatively small set of support vectors play a dominant role. This identification of support vectors is of interest when the number of training samples is large (simplifying subsequent prediction). The key reason to invoke a Bayesian form of the SVM [5], instead of applying the widely studied optimization-based SVM [4], is that the former may be readily integrated into sophisticated hierarchical models. As an example of that, we here consider discriminative factor modeling, in which the factor scores are employed within a nonlinear SVM. We demonstrate the advantage of this in our experiments, with nonlinear discriminative factor modeling for high-dimensional gene-expression data.

We present MCMC and expectation conditional maximization inference for the model. Conditional conjugacy of the hierarchical model yields simple and efficient computations. Hence, while the nonlinear SVM is significantly more flexible than its linear counterpart, computations are only modestly more complicated. Details on the computational approaches, insights on the characteristics of the model, and demonstration on real data constitute a third contribution of this paper.

## 2   Mixture Representation for SVMs

**Previous model for linear SVM**   Assume $N$ observations $\{\mathbf{x}_n, y_n\}_{n=1}^N$, where $\mathbf{x}_n \in \mathbb{R}^d$ is a feature vector and $y_n \in \{-1, 1\}$ is its label. The support vector machine (SVM) seeks to find a classification function $f(\mathbf{x})$ by solving a regularized learning problem

$$\text{argmin}_{f(\mathbf{x})} \left\{ \gamma \sum_{n=1}^N \max(1 - y_n f(\mathbf{x}_n), 0) + R(f(\mathbf{x})) \right\}, \tag{1}$$

where $\max(1 - y_n f(\mathbf{x}_n), 0)$ is the hinge loss, $R(f(\mathbf{x}))$ is a regularization term that controls the complexity of $f(\mathbf{x})$, and $\gamma$ is a tuning parameter controlling the tradeoff between error penalization and the complexity of the classification function. The decision boundary is defined as $\{\mathbf{x} : f(\mathbf{x}) = 0\}$ and $\text{sign}(f(\mathbf{x}))$ is the decision rule, classifying $\mathbf{x}$ as either $-1$ or $1$ [4].

Recently, [5] showed that for the linear classifier $f(\mathbf{x}) = \boldsymbol{\beta}^\top \mathbf{x}$, minimizing (1) is equivalent to estimating the mode of the pseudo-posterior of $\boldsymbol{\beta}$

$$p(\boldsymbol{\beta}|\mathbf{X}, \mathbf{y}, \gamma) \propto \prod_{n=1}^N L(y_n|\mathbf{x}_n, \boldsymbol{\beta}, \gamma) p(\boldsymbol{\beta}|\cdot), \tag{2}$$

where $\mathbf{y} = [y_1 \ldots y_N]^\top$, $\mathbf{X} = [\mathbf{x}_1 \ldots \mathbf{x}_N]$, $L(y_n|\mathbf{x}_n, \boldsymbol{\beta}, \gamma)$ is the pseudo-likelihood function, and $p(\boldsymbol{\beta}|\cdot)$ is the prior distribution for the vector of coefficients $\boldsymbol{\beta}$. Choosing $\boldsymbol{\beta}$ to maximize the log of (2) corresponds to (1), where the prior is associated with $R(f(\mathbf{x}))$. In [5] it was shown that $L(y_n|\mathbf{x}_n, \boldsymbol{\beta}, \gamma)$ admits a location-scale mixture of normals representation by introducing latent variables $\lambda_n$, such that

$$L(y_n|\mathbf{x}_n, \boldsymbol{\beta}, \gamma) = e^{-2\gamma \max(1 - y_n \boldsymbol{\beta}^\top \mathbf{x}_n, 0)} = \int_0^\infty \frac{\sqrt{\gamma}}{\sqrt{2\pi\lambda_n}} \exp\left( -\frac{(1 + \lambda_n - y_n \boldsymbol{\beta}^\top \mathbf{x}_n)^2}{2\gamma^{-1}\lambda_n} \right) d\lambda_n. \tag{3}$$

Expression (2) is termed a *pseudo*-posterior because its likelihood term is unnormalized with respect to $y_n$. Note that an improper flat prior is imposed on $\lambda_n$.

The original formulation of [5] has the tuning parameter $\gamma$ as part of the prior distribution of $\boldsymbol{\beta}$, while here in (3) it is included instead in the likelihood. This is done because $(i)$ it puts $\lambda_n$ and the regularization term $\gamma$ together, and $(ii)$ it allows more freedom in the choice of the prior for $\boldsymbol{\beta}$. Additionally, it has an interesting interpretation, in that the SVM loss function behaves like a global-local shrinkage distribution [9]. Specifically, $\gamma^{-1}$ corresponds to a "global" scaling of the variance, and $\lambda_n$ represents the "local" scaling for component $n$. The $\{\lambda_n\}$ define the *relative* variances for each of the $N$ data, and $\gamma^{-1}$ provides a global scaling.

One of the benefits of a Bayesian formulation for SVMs is that we can flexibly specify the behavior of $\boldsymbol{\beta}$ while being able to adaptively regularize it by specifying a prior $p(\gamma)$ as well. For instance, [5] gave three examples of prior distributions for $\boldsymbol{\beta}$: Gaussian, Laplace, and spike-slab.

We can extend the results of [5] to a slightly more general loss function, by imposing a proper prior for the latent variables $\lambda_n$. In particular, by specifying $\lambda_n \sim \text{Exp}(\gamma_0)$ and letting $u_n = 1 - y_n \boldsymbol{\beta}^\top \mathbf{x}_n$,

$$L(y_n|\mathbf{x}_n, \boldsymbol{\beta}, \gamma) = \int_0^\infty \frac{\gamma_0 \sqrt{\gamma}}{\sqrt{2\pi\lambda}} e^{-\frac{\gamma}{2}\frac{(u_n + \lambda_n)^2}{\lambda_n}} e^{-\gamma_0 \lambda_n} d\lambda_n = \frac{\gamma_0}{c} e^{-\gamma(c|u_n| + u_n)}, \tag{4}$$

where $c = \sqrt{1 + 2\gamma_0\gamma^{-1}} > 1$. The proof relies (see Supplementary Material) on the identity, $\int_0^\infty a(2\pi\lambda)^{-1/2}\exp\{-\frac{1}{2}(a^2\lambda + b^2\lambda^{-1})\}d\lambda = e^{-|ab|}$ [10]. From (4) we see that as $\gamma_0 \to 0$ we recover (3) by noting that $2\max(u_n, 0) = |u_n| + u_n$. In general we may use the prior $\lambda_n \sim \text{Ga}(a_\lambda, \gamma_0)$, with $a_\lambda = 1$ for the exponential distribution. In the next section we discuss other choices for $a_\lambda$. This means that the proposed likelihood is no longer equivalent to the hinge loss but to a more general loss, termed below a skewed Laplace distribution.

**Skewed Laplace distribution**  We can write the likelihood function in (4) in terms of $u_n$ as

$$L(u_n|\gamma, \gamma_0) = \int_0^\infty \mathcal{N}(u_n| - \lambda_n, \gamma^{-1}\lambda_n)\text{Exp}(\lambda_n|\gamma_0)d\lambda_n = \frac{\gamma_0}{c}\begin{cases} e^{-\gamma(c+1)u_n}, & \text{if } u_n \geq 0 \\ e^{-\gamma(c-1)|u_n|}, & \text{if } u_n < 0 \end{cases}, \quad (5)$$

which corresponds to a Laplace distribution, with negative skewness, denoted as $\text{sLa}(u_n|\gamma, \gamma_0)$. Unlike the density derived from the hinge loss ($\gamma_0 \to 0$), this density is properly normalized, thus it corresponds to a valid probability density function. For the special case $\gamma_0 = 0$, the integral diverges, hence the normalization constant does not exist, which stems from $\exp(-2\gamma\max(u_n, 0))$ being constant for $-\infty < u_n < 0$.

From (5) we see that $\text{sLa}(u_n|\gamma, \gamma_0)$ can be represented either as mixture of normals or mixture of exponentials. Other properties of the distribution, such as its moments, can be obtained using the results for general asymmetric Laplace distributions in [11]. Examining (5) we can gain some intuition about the behavior of the likelihood function for the classification problem: ($i$) When $y_n\boldsymbol{\beta}^\top\mathbf{x}_n = 1$, $\lambda_n = 0$ and $\mathbf{x}_n$ lies on the margin boundary. ($ii$) When $y_n\boldsymbol{\beta}^\top\mathbf{x}_n > 1$, $\mathbf{x}_n$ is correctly classified, outside the margin and $|1 - y_n\boldsymbol{\beta}^\top\mathbf{x}_n|$ is exponential with rate $\gamma(c - 1)$. ($iii$) $\mathbf{x}_n$ is correctly classified but lies inside the margin when $0 < y_n\boldsymbol{\beta}^\top\mathbf{x}_n < 1$, and $\mathbf{x}_n$ is misclassified when $y_n\boldsymbol{\beta}^\top\mathbf{x}_n < 0$. In both cases, $1 - y_n\boldsymbol{\beta}^\top\mathbf{x}_n$ is exponential with rate $\gamma(c + 1)$. ($iv$) Finally, if $y_n\boldsymbol{\beta}^\top\mathbf{x}_n = 0$, $\mathbf{x}_n$ lies on the decision boundary.

Since $c + 1 > c - 1$ for every $c > 1$, the distribution for case ($ii$) decays slower than the distribution for case ($iii$). Alternatively, in terms of the loss function, observations satisfying ($iii$) get more penalized than those satisfying ($ii$). In the limiting case, $\gamma_0 \to 0$ we have $c \to 1$, and case ($ii$) is not penalized at all, recovering the behavior of the hinge loss. In the SVM literature, an observation $\mathbf{x}_n$ is called a support vector if it satisfies cases ($i$) or ($iii$). In the latter case, $\lambda_n$ is the distance from $y_n\boldsymbol{\beta}^\top\mathbf{x}_n$ to the margin boundary [4]. The key thing that the $\text{Exp}(\lambda_0)$ prior imposes on $\lambda_n$, relative to the flat prior on $\lambda_n \in [0, \infty)$, is that it constrains that $\lambda_n$ not be too large (discouraging $y_n\boldsymbol{\beta}^\top\mathbf{x}_n \gg 1$ for correct classifications, which is even more relevant for nonlinear SVMs); we discuss this further below.

**Extension to nonlinear SVM**  We now assume that the decision function $f(\mathbf{x})$ is drawn from a zero-mean Gaussian process $\text{GP}(\mathbf{0}, k(\mathbf{x}, \cdot, \boldsymbol{\theta}))$, with kernel parameters $\boldsymbol{\theta}$. Evaluated at the $N$ points at which we have data, $\mathbf{f} \sim \mathcal{N}(\mathbf{0}, \mathbf{K})$, where $\mathbf{K}$ is a $N \times N$ covariance matrix with entries $k_{ij} = k(\mathbf{x}_i, \mathbf{x}_j, \boldsymbol{\theta})$ for $i, j \in \{1, \ldots, N\}$ [7]; $\mathbf{f} = [f_1 \ldots f_N]^\top \in \mathbb{R}^N$ corresponds to the continuous $f(\mathbf{x})$ evaluated at $\{\mathbf{x}_n\}_{n=1}^N$. Together with (5), for $u_n = 1 - y_n f_n$, where $f_n = f(\mathbf{x}_n)$, the full prior specification for the *nonlinear* SVM is

$$\mathbf{f} \sim \mathcal{N}(\mathbf{0}, \mathbf{K}), \quad \lambda_n \sim \text{Exp}(\gamma_0), \quad \gamma \sim \text{Ga}(a_0, b_0). \quad (6)$$

It is straightforward to prove the equality in (5) holds for $f_n$ in place of $\boldsymbol{\beta}^\top\mathbf{x}_n$, as in (6).

For nonlinear SVMs as above, being able to set $\gamma_0 > 0$ is particularly beneficial. It prevents $f_n$ from being arbitrarily large (hence preventing $1 - y_n f_n \ll 0$). This implies that isolated observations far away from linear decision boundary (even when correctly classified when learning) tend to be support vectors in a nonlinear SVM, yielding more conservative learned nonlinear decision boundaries. Figure 1 shows examples of $\log\mathcal{N}(1 - y_n f_n; -\lambda_n, \gamma^{-1}\lambda_n)\text{Exp}(\lambda_n; \gamma_0)$ for $\gamma = 100$ and $\gamma_0 = \{0.01, 100\}$. The vertical lines denote the margin boundary ($y_n f_n = 1$) and the decision boundary ($y_n f_n = 0$). We see that when $\gamma_0$ is small, the density has a very pronounced negative skewness (like in the hinge loss of the original SVM) whereas when $\gamma_0$ is large, the density tends to be more of a symmetric shape.

## 3  Inference

We wish to compute the posterior $p(\mathbf{f}, \boldsymbol{\lambda}, \gamma|\mathbf{y}, \mathbf{X})$, where $\boldsymbol{\lambda} = [\lambda_1 \ldots \lambda_N]^\top$. We describe and have implemented three inference procedures: Markov chain Monte Carlo (MCMC), a point estimate via expectation-conditional maximization (ECM) and a GP approximation for fast inference.

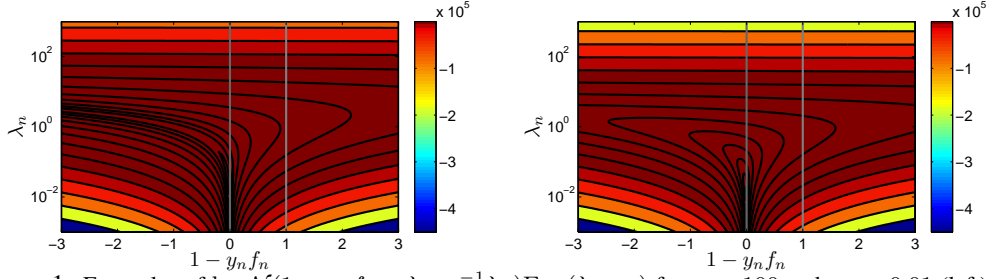

Figure 1: Examples of $\log \mathcal{N}(1 - y_n f_n; -\lambda_n, \gamma^{-1}\lambda_n)\mathrm{Exp}(\lambda_n; \gamma_0)$ for $\gamma = 100$ and $\gamma_0 = 0.01$ (left) and $\gamma_0 = 100$ (right). The vertical lines denote the margin boundary ($y_n f_n = 1$) and the decision boundary ($y_n f_n = 0$).

**MCMC** Inference is implemented by repeatedly sampling from the conditional posterior of parameters in (6). Conditional conjugacy allows us to express the following distributions in closed form:

$$\mathbf{f}|\mathbf{y}, \boldsymbol{\lambda}, \gamma \sim \mathcal{N}(\mathbf{m}, \mathbf{S}), \quad \mathbf{m} = \gamma \mathbf{S}\mathbf{Y}\boldsymbol{\Lambda}^{-1}(1 + \boldsymbol{\lambda}), \quad \mathbf{S} = \gamma^{-1}\mathbf{K}(\mathbf{K} + \gamma^{-1}\boldsymbol{\Lambda})^{-1}\boldsymbol{\Lambda},$$

$$\lambda_n^{-1}|f_n, y_n, \gamma \sim \mathrm{IG}\left(\frac{\sqrt{1 + 2\gamma_0\gamma^{-1}}}{|1 - y_n f_n|}, \gamma + 2\gamma_0\right), \quad \gamma|\mathbf{y}, \mathbf{f}, \boldsymbol{\lambda} \sim \mathrm{Ga}\left(a_0 + \frac{1}{2}N, b_0 + \frac{1}{2}\boldsymbol{\epsilon}^\top \boldsymbol{\Lambda}^{-1}\boldsymbol{\epsilon}\right), \quad (7)$$

where $\boldsymbol{\Lambda} = \mathrm{diag}(\boldsymbol{\lambda})$, $\mathbf{Y} = \mathrm{diag}(\mathbf{y})$, $\boldsymbol{\epsilon} = 1 + \boldsymbol{\lambda} - \mathbf{Yf}$, and $\mathrm{IG}(\mu, \gamma)$ is the inverse Gaussian distribution with parameters $\mu$ and $\gamma$ [10].

In MCMC $\gamma_0$ plays a crucial role, because it controls the prior variance of the latent variables $\lambda_n$, thus greatly improving mixing, particularly that of $\gamma$. We also verified empirically that for small values of $\gamma_0$, $\gamma$ is consistently underestimated. In practice we fix $\gamma_0 = 0.1$, however, a conjugate prior (gamma) exists, and sampling from its conditional posterior is straightforward if desired.

The parameters of the covariance function $\boldsymbol{\theta}$ in the GP require Metropolis-Hastings type algorithms, as in most cases no closed form for their conditional posterior is available. However, the problem is relatively well studied. We have found that slice sampling methods [12], in particular the surrogate data sampler of [13], work well in practice, and are employed here.

For the case of SVMs, MCMC is naturally important as a way of quantifying the uncertainty of the parameters of the model. Further, it allows us to use the hierarchy in (6) as a building block in more sophisticated models, or to bring more flexibility to $\mathbf{f}$ through specialized prior specifications. As an example of this, Section 5 describes a specification for a nonlinear discriminative factor model.

**ECM** The expectation-conditional maximization algorithm is a generalization of the expectation-maximization (EM) algorithm. It can be used when there are multiple parameters that need to be estimated [14]. From (6) we identify $\mathbf{f}$ and $\gamma$ as the parameters to be estimated, and $\lambda_n$ as the latent variables. The $Q$ function in EM-style algorithms is the complete data log-posterior, where expectations are taken w.r.t. the posterior distribution evaluated at the current value of the parameter of interest. From (7) we see that $\lambda_n$ appears in the conditional posterior $p(\mathbf{f}|\mathbf{y}, \mathbf{K}, \boldsymbol{\lambda}, \gamma)$ as first order terms, thus we can write

$$\langle \lambda_n^{-1}\rangle = \mathbb{E}[\lambda_n^{-1}|y_n, f_n^{(i)}, \gamma^{(i)}] = \sqrt{1 + 2\gamma_0(\gamma^{(i)})^{-1}}|u_n^{(i)}|^{-1}, \quad (8)$$

where $f_n^{(i)}$ and $\gamma^{(i)}$ are the estimates of $f_n$ and $\gamma$ at the $i$-th iteration, and $u_n^{(i)} = 1 - y_n f_n^{(i)}$. From (7) and (8) we can obtain the EM updates: $\mathbf{f}^{(i+1)} = \mathbf{K}(\mathbf{K} + (\gamma^{(i)})^{-1}\langle\boldsymbol{\Lambda}\rangle)^{-1}\mathbf{Y}(1 + \langle\boldsymbol{\lambda}\rangle)$ and

$$\gamma^{(i+1)} = \left(a_0 - 1 + \tfrac{1}{2}N\right)\left(b_0 + \tfrac{1}{2}\sum_{n=1}^N \langle\lambda_n^{-1}\rangle(u_n^{(i+1)})^2 + 2u_n^{(i+1)} + \langle\lambda_n\rangle\right)^{-1}.$$

In the ECM setting, learning the parameters of the covariance function is not as straightforward as in MCMC. However, we can borrow from the GP literature [7] and use the fact that we can marginalize $\mathbf{f}$ while conditioning on $\boldsymbol{\lambda}$ and $\gamma$:

$$Z(\mathbf{y}, \mathbf{X}, \boldsymbol{\lambda}, \gamma, \boldsymbol{\theta}) = \mathcal{N}(\mathbf{Y}(1 + \boldsymbol{\lambda}), \mathbf{K} + \gamma^{-1}\boldsymbol{\Lambda}). \quad (9)$$

Note that $\mathbf{K}$ is a function of $\mathbf{X}$ and $\boldsymbol{\theta}$. Estimation of $\boldsymbol{\theta}$ is done by maximizing $\log Z(\mathbf{y}, \mathbf{X}, \boldsymbol{\lambda}, \gamma, \boldsymbol{\theta})$. For this we need only compute the partial derivatives of (9) w.r.t. $\boldsymbol{\theta}$, and then use a gradient-based

optimizer. This is commonly known as Type II maximum likelihood (ML-II) [7]. In practice we alternate between EM updates for $\{\mathbf{f}, \gamma\}$ and $\boldsymbol{\theta}$ updates for a pre-specified number of iterations (typically the model converges after 20 iterations).

**Speeding up inference** Perhaps one of the most well known shortcomings of GP is that its cubic complexity is prohibitive for large scale problems. However there is an extensive literature about approximations for fast GP models [15]. Here we use the Fully Independent Training Conditional (FITC) approximation [16], as it offers an attractive balance between complexity and performance [15]. The basic idea behind FITC is to assume that $\mathbf{f}$ is generated i.i.d. from *pseudo-inputs* $\{\mathbf{v}_m\}_{m=1}^M$ via $\mathbf{f}_m \in \mathbb{R}^M$ such that $\mathbf{f}_m \sim \mathcal{N}(\mathbf{0}, \mathbf{K}_{mm})$, where $\mathbf{K}_{mm}$ is a $M \times M$ covariance matrix. Specifically, from (5) we have

$$p(\mathbf{u}|\mathbf{f}_m) = \prod_{n=1}^N p(u_n|\mathbf{f}_m) = \mathcal{N}(\mathbf{K}_{nm}\mathbf{K}_{mm}^{-1}\mathbf{f}_m, \mathrm{diag}(\mathbf{K} - \mathbf{Q}_{nn}) + \gamma^{-1}\boldsymbol{\Lambda}),$$

where $\mathbf{u} = 1 - \mathbf{Yf}$, $\mathbf{K}_{mn}$ is the cross-covariance matrix between $\{\mathbf{v}_m\}_{m=1}^M$ and $\{\mathbf{x}_n\}_{n=1}^N$, and $\mathbf{Q}_{nn} = \mathbf{K}_{nm}\mathbf{K}_{mm}^{-1}\mathbf{K}_{mn}$. If we marginalize out $\mathbf{f}_m$ thus

$$Z(\mathbf{y}, \mathbf{X}, \boldsymbol{\lambda}, \gamma, \boldsymbol{\theta}) = \mathcal{N}(\mathbf{Y}(1 + \boldsymbol{\lambda}), \mathbf{Q}_{nn} + \mathrm{diag}(\mathbf{K} - \mathbf{Q}_{nn}) + \gamma^{-1}\boldsymbol{\Lambda}). \tag{10}$$

Note that if we drop the $\mathrm{diag}(\cdot)$ term in (10) due to the i.i.d. assumption for $\mathbf{f}$, we recover the full GP marginal from (9). Similar to the ML-II approach previously described, for a fixed $M$ we can maximize $\log Z(\mathbf{y}, \mathbf{X}, \boldsymbol{\lambda}, \gamma, \boldsymbol{\theta})$ w.r.t. $\boldsymbol{\theta}$ and $\{\mathbf{v}_m\}_{m=1}^M$ using a gradient-based optimizer but with the added benefit of having decreased the computational cost from $\mathcal{O}(N^3)$ to $\mathcal{O}(NM^2)$ [16].

**Predictions** Making predictions under the model in (6), with conditional posterior distributions in (7), can be achieved using standard results of the multivariate normal distribution. The predictive distribution of $f_\star$ for a new observation $\mathbf{x}_\star$ given the dataset $\{\mathbf{X}, \mathbf{y}\}$ can be written as

$$f_\star|\mathbf{x}_\star, \mathbf{X}, \mathbf{y} \sim \mathcal{N}(\mathbf{k}_\star\boldsymbol{\Sigma}\mathbf{Y}(1 + \boldsymbol{\lambda}), k_\star - \mathbf{k}_\star^\top\boldsymbol{\Sigma}\mathbf{k}_\star), \tag{11}$$

where $\boldsymbol{\Sigma} = (\mathbf{K} + \gamma^{-1}\boldsymbol{\Lambda})^{-1}$, $k_\star = k(\mathbf{x}_\star, \mathbf{x}_\star, \boldsymbol{\theta})$ and $\mathbf{k}_\star = [k(\mathbf{x}_\star, \mathbf{x}_1, \boldsymbol{\theta}) \ \ldots \ k(\mathbf{x}_\star, \mathbf{x}_N, \boldsymbol{\theta})]^\top$. Furthermore, we can directly use the probit link $\Phi(f_\star)$ to compute

$$p(y_\star = 1|\mathbf{x}_\star, \mathbf{X}, \mathbf{y}) = \int \Phi(f_\star)p(f_\star|\mathbf{x}_\star, \mathbf{X}, \mathbf{y})df_\star = \Phi\left(\mathbf{k}_\star\boldsymbol{\Sigma}\mathbf{Y}(1 + \boldsymbol{\lambda})(1 + k_\star - \mathbf{k}_\star^\top\boldsymbol{\Sigma}\mathbf{k}_\star)^{-1}\right),$$

which follows from [7]. Computing the class membership probability is not possible in standard SVMs, because in such optimization-based methods one does not obtain the variance of the predictive distribution; this variance is an attractive component of the Bayesian construction.

The mean of the predictive distribution (11) is tightly related to the predictor in standard SVMs, in the sense that both are manifestations of the representer theorem. In particular

$$\mathbb{E}[f_\star|\mathbf{x}_\star, \mathbf{X}, \mathbf{y}] = \sum_{n=1}^N \alpha_n k(\mathbf{x}_\star, \mathbf{x}_n, \boldsymbol{\theta}), \tag{12}$$

where $\boldsymbol{\alpha} = (\mathbf{K} + \gamma^{-1}\boldsymbol{\Lambda})^{-1}\mathbf{Y}(1 + \boldsymbol{\lambda})$. From the expectations of $\lambda_n$ and $\mathbf{f}$ conditioned on $\gamma$ and $\gamma_0$ it is possible to show that $\boldsymbol{\alpha}$ is a vector with elements $\gamma(1 - c) \le \alpha_n \le \gamma(1 + c)$, where $c = \sqrt{1 + 2\gamma_0\gamma^{-1}}$. We differentiate three types of elements in $\boldsymbol{\alpha}$ as follows

$$\boldsymbol{\alpha} = \begin{cases} y_n\gamma(1 + c), & \text{if } y_nf_n < 1 \\ \alpha_n^0, & \text{if } y_nf_n = 1 \ (\lambda_n = 0) \\ y_n\gamma(1 - c), & \text{if } y_nf_n > 1 \end{cases}, \tag{13}$$

with $\boldsymbol{\alpha}_0 = \mathbf{K}_{0,0}^{-1}(\mathbf{y}_0 - \gamma(1 + c)\mathbf{K}_{0,a}\mathbf{y}_a - \gamma(1 - c)\mathbf{K}_{0,b}\mathbf{y}_b)$, where $\alpha_n^0$ is an element of $\boldsymbol{\alpha}_0$, and $0$, $a$ and $b$ are subsets of $\{1, \ldots, N\}$ for which $\lambda_n = 0$, $y_nf_n < 1$ and $y_nf_n > 1$, respectively. This implies $\boldsymbol{\alpha}$ and so the prediction rule in (12) depend on data for which $\lambda_n > 0$ only through $\gamma$ and $\gamma_0$. Note also that we do not need the values of $\boldsymbol{\lambda}$ but whether or not they are different than zero. When $\gamma_0 \to 0$ then $c \to 1$ and $\boldsymbol{\alpha}$ becomes a sparse vector bounded above by $2\gamma$. This result for standard SVMs can be found independently from the Karush-Kuhn-Tucker conditions for its objective function [4].

For ECM and variational Bayes EM inference (the latter discussed below in Section 5), we set $\gamma_0 = 0$ and therefore $\boldsymbol{\alpha}$ is sparse, with $\alpha_n = 0$ when $y_nf_n > 1$, as in traditional SVMs. This property of the proposed use of GPs within the Bayesian SVM formulation is a significant advantage relative to traditional classifier design based directly on GPs, for which we do not have such sparsity in general. For MCMC inference, we find the sampler mixes better when $\gamma_0 \neq 0$. Details on the derivations of (13) and the concavity of the problem may be found in Supplementary Material.

## 4 Related Work

A key contribution of this paper concerns extension of the *linear* Bayesian SVM developed in [5] to a *nonlinear* Bayesian SVM. This has been implemented by replacing the linear $f(\mathbf{x}) = \boldsymbol{\beta}^\top \mathbf{x}$ considered in [5] with an $f(\mathbf{x})$ drawn from a GP. The most relevant previous work is that for which a classifier is directly implemented via a GP, without an explicit connection to the margin associated with the SVM [7]. Specifically, GP-based classifiers have been developed by [17]. In [7] the $\mathbf{f}$ is drawn from a GP, as in (6), but $\mathbf{f}$ is used directly with a probit or logit link function, to estimate class membership probability. Previous GP-based classifiers did not use $\mathbf{f}$ within a margin-based classifier as in (6), implemented here via $p(u_n) = \mathcal{N}(-\lambda_n, \gamma^{-1}\lambda_n)$, where $u_n = 1 - y_n f_n$. It has been shown empirically that nonlinear SVMs and GP classifiers often perform similarly [8]. However, for the latter, inference can be challenging due to the non-conjugacy of multivariate normal distribution to the link function. Common inference strategies employ iterative approximate inference schemes, such as the Laplace approximation [17] or expectation propagation (EP) [18]. The model we propose here is locally fully conjugate (except for the GP kernel parameters) and inference can be easily implemented using EM style algorithms, or via MCMC. Besides, the prediction rule of the GP classifier, which has a form almost identical to (12), is generally not sparse and therefore lacks the interpretation that may be provided by the relatively few support vectors.

## 5 Discriminative Factor Models

Combinations of factor models and *linear* classifiers have been widely used in many applications, such as gene expression, proteomics and image analysis, as a way to perform classification and feature selection simultaneously [19, 20]. One of the most common modeling approaches can be written as
$$\mathbf{x}_n = \mathbf{A}\mathbf{w}_n + \boldsymbol{\epsilon}_n, \quad \boldsymbol{\epsilon}_n \sim \mathcal{N}(0, \psi^{-1}\mathbf{I}), \quad L(y_n|\boldsymbol{\beta}, \mathbf{w}_n, \cdot),$$

where $\mathbf{A}$ is a $d \times K$ matrix of factor loadings, $\mathbf{w}_n \in \mathbb{R}^K$ is a vector of factor scores, $\boldsymbol{\epsilon}_n$ is observation noise (and/or model residual), $\boldsymbol{\beta}$ is a vector of $K$ linear classifier coefficients and $L(\cdot)$ is for instance but not limited to the linear SVM likelihood in (5) (a logit or probit link may also be used). One of many possible prior specification for the above model is
$$\mathbf{a}_k \sim \mathcal{N}(\mathbf{0}, \boldsymbol{\Phi}_k), \quad \mathbf{w}_n \sim \mathcal{N}(\mathbf{0}, \mathbf{I}), \quad \psi \sim \text{Ga}(a_\psi, b_\psi), \quad \boldsymbol{\beta} \sim \mathcal{N}(\mathbf{0}, \mathbf{G}),$$

where $\mathbf{a}_k$ is a column of $\mathbf{A}$, $\boldsymbol{\Phi}_k = \text{diag}(\phi_{1k}, \dots, \phi_{dk})$, $\phi_{ik} \sim \text{Exp}(\nu)$, $\mathbf{G} = \text{diag}(g_1, \dots, g_K)$ and each element of $\mathbf{A}$ is distributed $a_{ik} \sim \text{Laplace}(\nu)$ after marginalizing out $\{\phi_{ik}\}$ [10]. Shrinkage in $\mathbf{A}$ is typically a requirement when $N \ll d$ or when its columns, $\mathbf{a}_k$, need to be interpreted. For simplicity, we can set $\mathbf{G} = \mathbf{I}$, however a shrinkage prior for the elements $g_k$ of $\mathbf{G}$ might be useful in some applications, as a mechanism for factor score selection. Although the described model usually works well in practice, it assumes that there is a linear mapping from $\mathbb{R}^d$ to $\mathbb{R}^K$, such that $K \ll d$, in which the classes $\{-1, 1\}$ are linearly separable. We can relax this assumption by imposing the hierarchical model in (6) in place of $\boldsymbol{\beta}$. This implies that matrix $\mathbf{K}$ from (6) has now entries $k_{ij} = k(\mathbf{w}_i, \mathbf{w}_j, \boldsymbol{\theta})$. Inference using MCMC is straightforward except for the conditional posterior of the factor scores. This model is related to latent-variable GP models (GP-LVM) [21], in that we infer the latent $\{\mathbf{w}_i\}$ that reside within a GP kernel. However, here $\{\mathbf{w}_i\}$ are also factor scores in a factor model, and the GP is used within the context of a Bayesian SVM classifier; neither of latter two have been considered previously.

For the nonlinear Bayesian SVM classifier we no longer have a closed form for the conditional of $\mathbf{w}_n$, due to the covariance function of the GP prior. Thus, we require a Metropolis-Hastings type algorithm. Here we use elliptical slice sampling [22]. Specifically, we sample $\mathbf{w}_n$ from
$$p(\mathbf{w}_n|\mathbf{A}, \mathbf{W}_{\setminus n}, \psi, \mathbf{y}, \boldsymbol{\lambda}, \gamma, \boldsymbol{\theta}) \propto p(\mathbf{w}_n|\mathbf{x}_n, \mathbf{A}, \psi) Z(\mathbf{y}, \mathbf{w}_n, \mathbf{W}_{\setminus n}, \boldsymbol{\lambda}, \gamma, \boldsymbol{\theta}), \tag{14}$$

where $p(\mathbf{w}_n|\mathbf{x}_n, \mathbf{A}, \psi) \sim \mathcal{N}(\mathbf{S}_\text{N}\psi\mathbf{A}\mathbf{x}_n, \mathbf{S}_\text{N})$, $\mathbf{W} = [\mathbf{w}_1 \ \dots \ \mathbf{w}_N]$, $\mathbf{W}_{\setminus n}$ is matrix $\mathbf{W}$ without column $n$, $\mathbf{S}_\text{N}^{-1} = \psi\mathbf{A}^\top\mathbf{A} + \mathbf{I}$, and we have marginalized out $\mathbf{f}$ as in (9) with $\mathbf{W}$ in place of $\mathbf{X}$. The elliptical slice sampler proposes samples from $p(\mathbf{w}_n|\mathbf{x}_n, \mathbf{A}, \psi)$ while biasing them towards more likely configurations of $\boldsymbol{\lambda}$. Provided that $\boldsymbol{\lambda}$ ultimately controls the predictive distribution of the classifier in (11), samples of $\mathbf{w}_n$ will at the same time attempt to fit the data and to improve classification performance. From (14), note that we sample one column of $\mathbf{W}$ at a time, while keeping the others fixed. Details of the elliptical slice sampler are found in [22]. In applications in which sampling from (14) is time prohibitive, we can use instead a variational Bayes EM (VB-EM) approach. In the E-step, we approximate the posterior of $\mathbf{A}$, $\{\boldsymbol{\Phi}_k\}$, $\psi$, $\mathbf{f}$, $\boldsymbol{\lambda}$ and $\gamma$ by a factorized distribution $q(\mathbf{A})\prod_k q(\boldsymbol{\Phi}_k)q(\psi)q(\mathbf{f})q(\boldsymbol{\lambda})q(\gamma)$ and in the M-step we optimize $\mathbf{W}$ and $\boldsymbol{\theta}$, using L-BFGS [23]. Details of the implementation can be found in the Supplementary Material.

## 6  Experiments

In all experiments we set the covariance function to ($i$) either the square exponential (SE), which has the form $k(\mathbf{x}_i, \mathbf{x}_j, \theta) = \exp\left(-\|\mathbf{x}_i - \mathbf{x}_j\|^2 / \theta^2\right)$, where $\theta^2$ is known as the characteristic length scale; or ($ii$) the automatic relevance determination (ARD) SE in which each dimension of $\mathbf{x}$ has its own length scale [7]. All code used in the experiments was written in Matlab and executed on a 2.8GHz workstation with 4Gb RAM.

**Benchmark data**  We first compare the performance of the proposed Bayesian hierarchy for nonlinear SVM (BSVM) against EP-based GP classification (GPC) and an optimization-based SVM, on six well known benchmark datasets. In particular, we use the same data and settings as [8], specifically 10-fold cross-validation and SE covariance function. The parameters of the SVM $\{\gamma, \theta\}$ are obtained by grid search using

Table 1: Benchmark data results.  Mean % error from 10-fold cross-validation.

| Data set | $N$ | $d$ | BSVM | SVM | GPC |
|---|---|---|---|---|---|
| Ionosphere | 351 | 34 | 5.98 | **5.71** | 7.41 |
| Sonar | 208 | 60 | **11.06** | 11.54 | 12.50 |
| Wisconsin | 683 | 9 | 2.93 | 3.07 | **2.64** |
| Crabs | 200 | 7 | **1.5** | 2.0 | 2.5 |
| Pima | 768 | 8 | **21.88** | 24.22 | 22.01 |
| USPS 3 vs 5 | 1540 | 256 | **1.49** | 1.56 | 1.69 |

an internal 5-fold cross-validation. GPC uses ML-II and a modified SE function $k(\mathbf{x}_i, \mathbf{x}_j, \boldsymbol{\theta}) = \theta_1^2 \exp\left(-\|\mathbf{x}_i - \mathbf{x}_j\|^2 / \theta_2^2\right)$, where $\theta_1$ acts as regularization trade-off similar to $\gamma$ in our formulation [7]. For our model we set 200 as the maximum number of iterations of the ECM algorithm and run ML-II every 20 iterations. Table 1 shows mean errors for the methods under consideration. We see that all three perform similarly as one might expect thus error bars are not showed, however BSVM slightly outperforms the others in 4 out of 6 datasets. From the three methods, the SVM is clearly faster than the others. GP classification and our model essentially scale cubically with $N$, however, ours is relatively faster mainly due to overhead computations needed by the EP algorithm. More specifically, running times for the larger dataset (USPS 3 vs 5) were approximately 1000, 1200 and 5000 seconds for SVM, BSVM and GPC, respectively.

Table 2: FITC results (mean % error) for USPS data.

| | 3 vs. 5 ($N = 767$) | | 4 vs. non-4 ($N = 7291$) | |
|---|---|---|---|---|
| | FITC-GPC | FITC-BSVM | FITC-GPC | FITC-BSVM |
| Error | $3.69 \pm 0.26$ | $\mathbf{3.49 \pm 0.29}$ | $2.59 \pm 0.17$ | $\mathbf{2.44 \pm 0.17}$ |
| Time | 102 | 46 | 604 | 116 |

In order to test the approximation introduced in Section 3 (to accelerate GP inference) we use the traditional splitting of USPS, 7291 for model fitting and the remaining 2007 for testing, on two different tasks: 3 vs. 5 and 4 vs. non-4. Table 2 shows mean error rates and standard deviations for FITC versions of BSVM and GPC, for $M = 100$ pseudo-inputs and 10 repetitions. We see that FITC-BSVM slightly outperforms FITC-GPC in both tasks while being relatively faster. As baselines, full BSVM and GPC on the 3 vs. 5 task perform roughly similar at 2.46% error. We also verified (results not shown) that increasing $M$ consistently decreases error rates for both FITC-BSVM and FITC-GPC.

**USPS data**  We applied the model proposed in Section 5 to the well known 3 vs. 5 subset of the USPS handwritten digits dataset, consisting of 1540 gray scale $16 \times 16$ images, rescaled within $[-1, 1]$. We use the resampled version, this is, 767 for model fitting and the remaining 773 for testing. As baselines, we also perform inference as a two step procedure, first fitting the factor model (FM), followed by a linear (L) or a nonlinear (N) SVM classifier. We also consider learning jointly the factor model but with a linear SVM (LDFM), and a two step procedure consisting of LDFM followed by a nonlinear SVM. Our proposed nonlinear discriminative factor model is denoted NDFM. VB-EM versions of LDFM and NDFM are denoted as VLDFM and VNDFM, respectively. MCMC details for the linear SVM part can be found in [5]. For inference, we set $K = 10$, a SE covariance function and run the sampler for 1200 iterations, from which we discard the first 600 and keep every 10-th for posterior summaries. We observed in general good mixing regardless of random initialization, and results remained very similar for different Markov chains.

Table 3 shows classification results for the eight classifiers considered; we see that the nonlinear classifiers perform substantially better than the linear counterparts. In addition, the proposed nonlinear joint model (NDFM) is the best of all five. The nonlinear classifier is powerful enough to perform well in both two step procedures. We found that VNDFM is not performing as good as NDFM because the data likelihood is dominating over the labels likelihood in the updates for the factor scores, which is not surprising considering the marked size differences between the two. On the positive side, runtime for VNDFM is approximately two orders of magnitude smaller than that of NDFM. We also tried a joint nonlinear model with a probit link as in GP classification and we

Table 3: Mean % error with standard deviations and runtime (seconds) for USPS and gene expression data.

| | FM+L | FM+N | LDFM | VLDFM | LDFM+N | VLDFM+N | NDFM | VNDFM |
|---|---|---|---|---|---|---|---|---|
| | USPS (Test set) | | | | | | | |
| Error | $6.21 \pm 0.32$ | $3.36 \pm 0.26$ | $5.95 \pm 0.31$ | $5.56 \pm 0.18$ | $3.62 \pm 0.26$ | $3.62 \pm 0.19$ | $\mathbf{2.72 \pm 0.13}$ | $3.23 \pm 0.16$ |
| Time | 44 | 840 | 120 | 60 | 920 | 160 | 20000 | 210 |
| | Gene expression (10-fold cross-validation) | | | | | | | |
| Error | $22.70 \pm 0.92$ | $19.52 \pm 1.02$ | $22.70 \pm 0.92$ | $22.31 \pm 0.78$ | $20.31 \pm 0.88$ | $19.52 \pm 0.88$ | $\mathbf{18.33 \pm 0.84}$ | $\mathbf{18.33 \pm 0.84}$ |
| Time | 105 | 136 | 126 | 25 | 158 | 57 | 1100 | 103 |

found its classification performance (a mean error rate of 3.10%) being slightly worse than that for NDFM. In addition, we found that using ARD SE covariance functions to automatically select for features of $\mathbf{A}$ and larger values of $K$ did not substantial changed the results.

**Gene expression data** The dataset originally introduced in [24] consists of gene expression measurements from primary breast tumor samples for a study focused towards finding expression patterns potentially related to mutations of the p53 gene. The original data were normalized using RMA and filtered to exclude genes showing trivial variation. The final dataset consists of 251 samples and 2995 normalized gene expression values. The labeling variable indicates whether or not a sample exhibits the mutation. We use the same baseline and inference settings from our previous experiment, but validation is done by 10-fold cross-validation. In preliminary results we found that factor score selection improves results, hence for the linear classifier (L) we used an exponential prior for the variances of $\boldsymbol{\beta}$, $g_k \sim \mathrm{Exp}(\rho)$, and for the nonlinear case (N) we set an ARD SE covariance function for $\mathbf{K}$. Table 3 summarizes the results, the nonlinear variants outperform their linear counterparts and our joint model perform slightly better than the others. Additionally, the joint nonlinear model with GP and probit link yielded an error rate of 19.52%.

As a way of quantifying whether the features (factor loadings) produced by FM, LDFM and NDFM are meaningful from a biological point of view, we performed Gene Ontology (GO) searches for the gene lists encoded by each column of $\mathbf{A}$. In order to quantify the strength of the association between GO annotations and our gene lists we obtained Bonferroni corrected $p$-values [25]. We thresholded the elements of matrix $\mathbf{A}$ such that $|a_{ik}| > 0.1$. Using the 10 lists from each model we found that FM, LDFM and NDFM produced respectively 5, 5 and 8 factors significantly associated to GO terms relevant to breast cancer. The GO terms are: fatty acid metabolism, induction of programmed cell death (apoptosis), anti-apoptosis, regulation of cell cycle, positive regulation of cell cycle, cell cycle and Wnt signaling pathway. The strongest associations in all models are unsurprisingly apoptosis and positive regulation of cell cycle, however, only NDFM produced a significant association to anti-apoptosis which we believe is responsible for the edge in performance of NDFM in Table 3.

## 7 Conclusion

We have introduced a fully Bayesian version of nonlinear SVMs, extending the previous restriction to linear SVMs [5]. Almost all of the existing joint feature-learning and classifier-design models assumed linear classifiers [2, 3, 26]. We have demonstrated in our experiments that there is a substantial performance improvement manifested by the nonlinear classifier. In addition, we have extended the Bayesian equivalent of the hinge loss to a more general loss function, for both linear and nonlinear classifiers. We have demonstrated that this approach enhances modeling flexibility, and yields improved MCMC mixing. The Bayesian setup allows one to directly compute class membership probabilities. We showed how to use the nonlinear SVM as a module in a larger model, and presented compelling results to highlight its potential. Point estimate inference using ECM is conceptually simpler and easier to implement than MCMC or GP classification, although MCMC is attractive for integrating the factor model and classifier (for example). We showed how FITC and VB-EM based approximations can be used in conjunction with the SVM nonlinear classifier and discriminative factor modeling, respectively, as a way to scale inference in a principled way.

**Acknowledgments**

The research reported here was funded in part by ARO, DARPA, DOE, NGA and ONR.

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
