[Supplementary Material]

# Bayesian Nonlinear Support Vector Machines and Discriminative Factor Modeling

**Ricardo Henao, Xin Yuan and Lawrence Carin**
Department of Electrical and Computer Engineering
Duke University, Durham, NC 27708
{r.henao,xin.yuan,lcarin}@duke.edu

## Skewed Laplace distribution

Defining $u_n = 1 - y_n \boldsymbol{\beta}^\top \mathbf{x}_n$ we can write

$$L(u_n|\gamma, \gamma_0) = \int_0^\infty \mathcal{N}(u_n| - \lambda_n, \gamma^{-1}\lambda_n) \mathrm{Exp}(\lambda_n|\gamma_0) d\lambda_n$$

$$= \frac{\gamma_0}{c} e^{-\gamma(c|u_n|+u_n)} = \frac{\gamma_0}{c} \begin{cases} e^{-\gamma(c+1)u_n}, & \text{if } u_n \geq 0 \\ e^{-\gamma(c-1)|u_n|}, & \text{if } u_n < 0 \end{cases}, \tag{1}$$

where $c = \sqrt{1 + 2\gamma_0\gamma^{-1}} > 1$.

We can rewrite the integral in (1) as

$$L(u_n|\gamma, \gamma_0) = \int_0^\infty \frac{\gamma_0\sqrt{\gamma}}{\sqrt{2\pi\lambda_n}} e^{-\frac{\gamma}{2}\frac{(u_n+\lambda_n)^2}{\lambda_n}} e^{-\gamma_0\lambda_n} d\lambda_n = \int_0^\infty \frac{\gamma_0\sqrt{\gamma}}{\sqrt{2\pi\lambda_n}} e^{-\frac{\gamma}{2}(u_n^2\lambda_n^{-1}+c^2\lambda_n)} e^{-\gamma u_n} d\lambda_n. \tag{2}$$

Using the identity [1]

$$\int_0^\infty \frac{a}{\sqrt{2\pi\lambda}} e^{-\frac{1}{2}(b^2\lambda^{-1}+a^2\lambda)} d\lambda = e^{-|ab|},$$

we can see that by making $b^2 = \gamma u_n^2$, $a^2 = \gamma c^2$ and multiplying through by $c^{-1}$, (2) reduces to

$$L(u_n|\gamma, \gamma_0) = \frac{\gamma_0}{c} e^{-\gamma c|u_n| - \gamma u_n}.$$

Verifying that (1) integrates to one can be seen from

$$\int_{-\infty}^\infty e^{-\gamma(c|u_n|+u_n)} du_n = \int_{-\infty}^0 e^{-\gamma(1-c)u_n} du_n + \int_0^\infty e^{-\gamma(c+1)u_n} du_n = \frac{1}{\gamma(c-1)} + \frac{1}{\gamma(c+1)} = \frac{c}{\gamma_0}.$$

## Support vectors

We can write the posterior of parameters $\mathbf{f}$ and $\boldsymbol{\lambda}$ as

$$p(\mathbf{f}, \boldsymbol{\lambda}|\mathbf{K}, \gamma, \gamma_0) \propto p(\mathbf{f}|\mathbf{K}) \prod_{n=1}^N L(y_n|\mathbf{f}_n, \lambda_n, \gamma) p(\lambda_n|\gamma_0).$$

The maximum a posteriori solution can be obtained as

$$\underset{\mathbf{f},\boldsymbol{\lambda}}{\mathrm{argmax}} \underbrace{\log p(\mathbf{f}|\mathbf{K}) + \sum_{n=1}^N \log L(y_n|\mathbf{f}_n, \lambda_n, \gamma) p(\lambda_n|\gamma_0)}_{H(\mathbf{f},\boldsymbol{\lambda})}.$$

Solving for $\lambda_n$ and $\mathbf{f}$ with prior $\lambda_n \sim \text{Ga}(3/2, \gamma_0)$ we have

$$\frac{\partial H(\mathbf{f}, \lambda_n)}{\partial \lambda_n} = 0, \Rightarrow \lambda_n = \frac{|1 - y_n f_n|}{\sqrt{1 + 2\gamma_0 \gamma^{-1}}} \tag{3}$$

$$\frac{\partial H(\mathbf{f}, \lambda_n)}{\partial \mathbf{f}} = 0, \Rightarrow \mathbf{f} = \mathbf{K}\boldsymbol{\alpha}, \tag{4}$$

where $\boldsymbol{\alpha} = (\mathbf{K} + \gamma^{-1}\boldsymbol{\Lambda})^{-1}\mathbf{Y}(1 + \boldsymbol{\lambda})$. Note that (3) and (4) are means of the conditional posterior of $\lambda_n$ and $\mathbf{f}$, respectively. We can rewrite $\boldsymbol{\alpha}$ as

$$\begin{bmatrix} \mathbf{K}_{\backslash n, \backslash n} + \gamma^{-1}\boldsymbol{\Lambda}_{\backslash n, \backslash n} & \mathbf{k}_{\backslash n, n} \\ \mathbf{k}_{n, \backslash n} & k_{n,n} + \gamma^{-1}\lambda_n \end{bmatrix} \begin{bmatrix} \boldsymbol{\alpha}_{\backslash n} \\ \alpha_n \end{bmatrix} = \begin{bmatrix} \mathbf{Y}_{\backslash n, \backslash n}(1 + \boldsymbol{\lambda}_{\backslash n}) \\ y_n(1 + \lambda_n) \end{bmatrix},$$

where we have split $\boldsymbol{\alpha}$ in two blocks, $\boldsymbol{\alpha}_{\backslash n}$ and $\alpha_n$ of size $N - 1$ and 1, respectively. For $\alpha_n$ we have

$$\alpha_n = (k_{n,n} + \gamma^{-1}\lambda_n)^{-1}(y_n(1 + \lambda_n) - \mathbf{k}_{n, \backslash n}\boldsymbol{\alpha}_{\backslash n}). \tag{5}$$

From (4) we also have

$$f_n = k_{n,n}\alpha_n + \mathbf{k}_{n, \backslash n}\boldsymbol{\alpha}_{\backslash n}. \tag{6}$$

From (3) we can see that

$$f_n = \begin{cases} y_n(1 + c\lambda_n) & \text{if } y_n f_n > 1 \\ y_n & \text{if } y_n f_n = 1 \ (\lambda_n = 0) \\ y_n(1 - c\lambda_n) & \text{if } y_n f_n < 1 \end{cases}. \tag{7}$$

where $c = \sqrt{1 + 2\gamma_0\gamma^{-1}} > 1$.

Replacing (7) and (6) in (5) we have

$$\boldsymbol{\alpha} = \begin{cases} y_n\gamma(1 + c), & \text{if } y_n f_n < 1 \\ \alpha_n^0, & \text{if } y_n f_n = 1 \ (\lambda_n = 0) \\ y_n\gamma(1 - c), & \text{if } y_n f_n > 1 \end{cases}, \tag{8}$$

with

$$\boldsymbol{\alpha}_0 = \mathbf{K}_{0,0}^{-1}(\mathbf{y}_0 - \gamma(1 + c)\mathbf{K}_{0,a}\mathbf{y}_a - \gamma(1 - c)\mathbf{K}_{0,b}\mathbf{y}_b),$$

where $\alpha_n^0$ is an element of $\boldsymbol{\alpha}_0$, and 0, $a$ and $b$ are subsets of $\{1, \ldots, N\}$ for which $\lambda_n = 0$, $y_n f_n < 1$ and $y_n f_n > 1$, respectively.

Provided that the mode of the conditional posterior of $\lambda_n$ for $\lambda_n \sim \text{Ga}(3/2, \gamma_0)$ matches the mean of the conditional posterior of $\lambda_n$ for $\lambda_n \sim \text{Exp}(\gamma_0)$, $\boldsymbol{\alpha}$ as in (8) also holds for the latter scenario because

$$\mathbb{E}[\lambda_n^{-1}|y_n, f_n, \gamma] = \frac{\sqrt{1 + 2\gamma_0\gamma^{-1}}}{|1 - y_n f_n|},$$

as in (3).

## Convexity of $-H(\boldsymbol{\lambda}, \mathbf{f})$

The Hessian matrix of $-H(\boldsymbol{\lambda}, \mathbf{f})$ can be written as

$$\mathbf{H} = \begin{bmatrix} \mathbf{A} & \mathbf{B} \\ \mathbf{B} & \mathbf{C} \end{bmatrix},$$

where $\mathbf{A} = \mathbf{K}^{-1} + \gamma\boldsymbol{\Lambda}^{-1}$, and $\mathbf{B}$ and $\mathbf{C}$ are diagonal matrices with elements $b_n = \gamma y_n(1 - y_n f_n)\lambda_n^{-2}$ and $c_n = \gamma(1 - y_n f_n)^2\lambda_n^{-3}$, respectively. From the Schur complement condition, we have that $\mathbf{H}$ is positive semidefinite (PSD) if both $\mathbf{A}$ and

$$\mathbf{U} = \mathbf{C} - \mathbf{B}\mathbf{A}^{-1}\mathbf{B},$$

are PSD. Since $\mathbf{K}$ and $\boldsymbol{\Lambda}$ are PSD, $\mathbf{A}$ is as well. We need to show that $\mathbf{U}$ is PSD. We can rewrite $\mathbf{U}$ as

$$\mathbf{U} = \gamma \mathbf{D} \left( \boldsymbol{\Lambda} - \gamma (\mathbf{K}^{-1} + \gamma \boldsymbol{\Lambda}^{-1})^{-1} \right) \mathbf{D} \,,$$

where $\mathbf{D} = \boldsymbol{\Lambda}^{-1}(\mathbf{I} - \mathbf{YF})\boldsymbol{\Lambda}^{-1}$, $\mathbf{Y} = \operatorname{diag}(\mathbf{y})$ and $\mathbf{F} = \operatorname{diag}(\mathbf{f})$. Because $\mathbf{DD}$ is diagonal with elements $d_i^2 \geq 0$, we only have to show that

$$\mathbf{G} = \boldsymbol{\Lambda} - \gamma (\mathbf{K}^{-1} + \gamma \boldsymbol{\Lambda}^{-1})^{-1} = (\boldsymbol{\Lambda}^{-1} + \gamma \boldsymbol{\Lambda}^{-1} \mathbf{K} \boldsymbol{\Lambda}^{-1})^{-1} \,, \tag{9}$$

where we have applied the matrix inversion lemma, is PSD.

Since $\mathbf{K}$ in (9) is PSD, $\mathbf{G}$, $\mathbf{U}$ and $\mathbf{H}$ are too, thus the negative log-posterior $-H(\mathbf{f}, \boldsymbol{\lambda})$ is convex.

## Fast inference for discriminative factor model

We use variational Bayes EM (VB-EM) approach. In the E-step, we approximate the posterior of $\mathbf{A}$, $\{\boldsymbol{\Phi}_k\}$, $\psi$, $\mathbf{f}$, $\boldsymbol{\lambda}$ and $\gamma$ by a factorized distribution $q(\mathbf{A}) \prod_k q(\boldsymbol{\Phi}_k) q(\psi) q(\mathbf{f}) q(\boldsymbol{\lambda}) q(\gamma)$ and in the M-step we optimize $\mathbf{W}$ and $\boldsymbol{\theta}$, using L-BFGS [2].

The goal is to minimize the Kullback-Leibler divergence between our factorized approximation and the exact posterior, to do so, we use coordinate ascent, i.e. we update one group of parameters at the time while keeping the remaining ones fixed. The inference algorithm iteratively cycles through updates for all parameters of the model. Updates for $\mathbf{A}$, $\boldsymbol{\Phi}_k$, $\psi$, $\lambda$ and $\gamma$ we can write

$$q(a_{ik}|-) = \mathcal{N}\left( c_{ik} \langle \psi \rangle \sum_{n=1}^{N} g_{\backslash in} w_{kn}, c_{ik} \right) \,,$$

$$q(\phi_{ik}^{-1}|-) = \mathrm{IG}\left( \sqrt{\frac{\nu}{\langle a_{ik}^2 \rangle}}, \nu \right) \,,$$

$$q(\psi|-) = \mathrm{Ga}\left( a_\psi + \frac{1}{2} dN, b_\psi + \frac{1}{2}\mathrm{tr}(\mathbf{X}\mathbf{X}^\top) - \mathrm{tr}(\mathbf{X}^\top \langle \mathbf{A} \rangle \mathbf{W}) + \frac{1}{2}\mathrm{tr}(\langle \mathbf{A}^\top \mathbf{A} \rangle \mathbf{W}\mathbf{W}^\top) \right) \,,$$

$$q(\mathbf{f}|-) = \mathcal{N}(\langle \gamma \rangle \mathbf{S} \mathbf{Y}(1 + \langle \boldsymbol{\lambda}^{-1} \rangle), \mathbf{S}) \,,$$

$$q(\lambda_n^{-1}|-) = \mathrm{IG}\left( \sqrt{\frac{\langle \gamma \rangle + 2\gamma_0}{\langle \gamma \rangle (1 - 2y_n \langle f_n \rangle + \langle f_n^2 \rangle)}}, \langle \gamma \rangle + 2\gamma_0 \right) \,,$$

$$q(\gamma|-) = \mathrm{Ga}\left( a_0 + \frac{1}{2}N, b_0 + \sum_{n=1}^{N} \frac{1}{2}\langle \lambda_n^{-1} \rangle (1 - 2y_n \langle f_n \rangle + \langle f_n^2 \rangle) + 1 - y_n \langle f_n \rangle + \frac{1}{2}\langle \lambda_n \rangle \right) \,,$$

where

$$\mathbf{G}_{\backslash in} = \mathbf{X} - \langle \mathbf{A} \rangle \mathbf{W} + \mathbf{a}_i \mathbf{w}_n \,, \quad c_{ik} = \langle \phi_{ik}^{-1} \rangle + \langle \psi \rangle \sum_{n=1}^{N} w_{kn}^2 \,, \quad \mathbf{S} = (\mathbf{K}^{-1} + \langle \gamma \rangle \langle \boldsymbol{\Lambda}^{-1} \rangle)^{-1} \,,$$

and $a_{ik}$, $\phi_{ik}$, $g_{\backslash in}$ and $f_n$ are elements of $\mathbf{A}$, $\boldsymbol{\Phi}_k$, $\mathbf{G}_{\backslash in}$ and $\mathbf{f}$, respectively.

We cannot obtain a closed form conditional distribution for the factor scores, $\mathbf{W}$, thus we optimize it by maximizing the following variational lower bound:

$$\mathcal{L}(\mathbf{W}) = \langle \psi \rangle \mathrm{tr}(\mathbf{X}^\top \langle \mathbf{A} \rangle \mathbf{W}) - \frac{1}{2}\langle \psi \rangle \mathrm{tr}(\langle \mathbf{A}^\top \mathbf{A} \rangle \mathbf{W}\mathbf{W}^\top)$$

$$- \frac{1}{2}\log \mathbf{U} - \frac{1}{2}\mathrm{tr}\left( \mathbf{U}^{-1}(\mathbf{I} + 2\langle \boldsymbol{\Lambda} \rangle + \langle \boldsymbol{\lambda}\boldsymbol{\lambda}^\top \rangle) \right)$$

$$- \frac{1}{2}\mathrm{tr}(\mathbf{W}\mathbf{W}^\top) + \text{const.} \,,$$

where $\mathbf{U} = \mathbf{K} + \langle \gamma^{-1} \rangle \langle \boldsymbol{\Lambda} \rangle$, $\langle \cdot \rangle$ denotes expectation and "const." encapsulates the terms not depending of $\mathbf{W}$. The gradient of $\mathbf{W}$ w.r.t. to $\mathcal{L}(\mathbf{W})$ can be written as

$$\frac{\partial \mathcal{L}}{\partial \mathbf{W}} = \langle \psi \rangle \langle \mathbf{A} \rangle^\top \mathbf{X} - \langle \psi \rangle \langle \mathbf{A}^\top \mathbf{A} \rangle \mathbf{W} - \frac{1}{2}\left\{ \mathbf{U}^{-1} - \mathbf{U}^{-1}(\mathbf{I} + 2\langle \boldsymbol{\Lambda} \rangle + \langle \boldsymbol{\lambda}\boldsymbol{\lambda}^\top \rangle)\mathbf{U}^{-1} \right\} \frac{\partial \mathbf{U}}{\partial \mathbf{W}} - \mathbf{W} \,,$$

where $\frac{\partial \mathbf{U}}{\partial \mathbf{W}}$ contains the derivatives of $\mathbf{W}$ w.r.t. $\mathbf{K}$ and it depends of $k(\mathbf{w}_i, \mathbf{w}_j, \boldsymbol{\theta})$.