[Reviews · NeurIPS 2014]

Submitted by Assigned_Reviewer_19

The authors "kernelize" a recent Bayesian SVM by use of Gaussian Processes. For the resulting hierarchical GP model two forms of approximate inference are proposed: An MCMC sampler and an EM algorithm.
A sparse (FITC) approximation is proposed as a further speedup.
Another contribution is a GPLVM model for the Bayesian kernel SVM.

In most places the paper is fairly clear in terms of writing.

Conceptually, most of the proposed extensions to the Bayesian SVM are pretty standard. But together they yield an interesting set of contributions.
Due to the large number of different contributions and extensions, each of these falls a bit short in terms of experiments. Despite assessing prediction quality on a range of data sets, some additional empirical validation beyond bare prediction errors and timing would be nice to assess properties of the model in terms of modeling and inference.
Summary: The topic and the proposed Bayesian kernel SVM are interesting and fairly relevant. I would have wished for some better intuition and empirical assessment of the properties of the model and differences to related models.

Submitted by Assigned_Reviewer_24

The author(s) present a method for combining the Bayes/GP and max-margin approaches to learning, initially by extending earlier work of [5]. They develop a detailed model combining Bayesian methods and SVM-related methods in a way that is clear, comprehensive and novel. They obtain algorithms for inference and prediction based on several statistical methods, including an algorithm incorporating a factor model, and give extensive experimental results.

Quality
-------

This is an excellent paper, presenting a new and apparently powerful collection of algorithms which seamlessly and elegantly integrate several state-of-the-art methods. While attempts to produce Bayesian versions of the SVM have a considerable history --- and I was somewhat surprised that no mention was made of the work of Peter Sollich et al., which I believe was the earliest --- this seems a cleaner and more satisfactory approach than those I have previously seen.

Clarity
-------

The paper is clear, concise and well-written.

Originality
-----------

The work demonstrates considerable originality.

Significance
------------

Summary: A very good paper indeed.

Submitted by Assigned_Reviewer_36

This paper mainly extends the Bayesian linear SVM in [5] to a nonlinear version, and further combines the Bayesian nonlinear SVM with factor models.
The extension from linear to nonlinear however is quite trivial by simple adopting the standard kernel tricks. The resulted nonlinear version involves more complicated inference problem since it will also learn the kernel function parameters.
The combination with factor models is produced by taking the two objectives together, while kernels are produced on the factor representations.

There is not much novelity in terms of model extension and combination strategies. The overall learning problem is in fact a quite complicated non-convex optimization problem.
Under the probabilistic Bayesian framework, some inference procedures are introduced to perform learning but there is no analysis about the complexity of the overall inference procedure.

The experiments are limited to using Gaussian kernels. Is it possible to use other types of nonlinear kernels? Will it affect the inference algorithm?

The datasets used in the experiments are too small (see Table 1). Large scale experiments need to be conducted.
Moreover, the authors only compared the proposed approach to SVM and GPC methods.
Considering the tasks addressed in this paper are simple binary classification tasks, why not compare to more advanced state-of-art methods?

The authors motivate the work from the perspective of discriminative feature-learning models, which is a very general topic.
I do not feel related works on this topic have been sufficiently discussed in the related work section.
Summary: This paper extends previous works from linear to nonlinear models.
The experiments are insufficient to demonstrate the efficacy of the approach.

Submitted by Assigned_Reviewer_41

This paper proposes an extension of the previous Bayesian formulation of conventional Support Vector Machines (SVMs). The previous Bayesian formulation proposed a likelihood of the form of Gaussian mixture, and the authors extend the formulation using the priors on the scale-mean parameter. This formulation results in a skewed Laplace likelihood which is different from the previous hinge loss. In addition, the Gaussian process extension of the Gaussian prior on the weight vectors presents a nonlinear SVM formulation. Two different optimization method is proposed for optimization.

The paper shows a nice example of extending the conventional SVM framework to a Bayesian setting using a well-defined formulation with prior distributions. According to the authors, the previous work [4] showed a connection between an infinite Gaussian mixture formulation and a hinge loss likelihood producing the SVM solution, but the formulation was improper due to the flat prior. The authors show how the formulation can be extended and how the inference can be performed using a well-defined formulation. The results show how the likelihood changed from the hinge-type likelihood to the skewed Laplacian. The explanation is clear and I enjoyed reading the manuscript. One thing I did not understand is the derivation of the predictive distribution in Eq. (11). Does the supplementary material include the derivation? I want a detailed derivation of Eq. (11) either in the paper or in the supplementary if the paper is accepted.
Summary: The paper is in general clear and the proposed formulation is correct. This paper can be a nice material for most of NIPS readers showing an example of Bayesian formulation of SVM.
Author Feedback
Author rebuttal: The paper presents several novel ideas: (i) It shows that the pseudo-likelihood function corresponding to the hinge-loss can be properly normalized and provides a geometric interpretation for the prior distribution (the skewed Laplace distribution) that gives rise to the normalized likelihood. (ii) It shows that Gaussian process classifiers can be used within a max-margin paradigm via variable augmentation in a conceptually simple manner. (iii) It shows how to scale inference in a principled way using pseudo-inputs. (iv) It provides closed form predictive distribution and a-posteriori class membership probabilities without ad-hoc procedures. The details of the derivation (to Reviewer #3) of the predictive distribution will be included in the supplementary material if the paper is accepted. Herein, we point to the references [7] and [15] due to the limited space. (v) It shows how to obtain well known expressions concerning the prediction rule and support vectors directly from the probabilistic formulation of SVMs, including the concavity of the objective function. (vi) It introduces a nonlinear discriminative factor model based on SVMs that can be readily scaled via variational Bayes EM.

The experiments are limited to squared exponential kernels. However, other types of kernels can be used without considerable additional effort, by leveraging existing work for Gaussian process based models.

The main purpose of the first experiment is to show that the proposed model performs as good as traditional SVMs and Gaussian process classifiers (GPCs) while retaining some of the best properties of both approaches, i.e., max-margin, sparsity and Bayesian hyperparameter selection. Although there are more advanced classification methods with state-of-the-art performance, we consider that SVMs and GPCs are still strong baselines and widely used classifiers in applications from many different domains. Besides, it is important to bear in mind that our goal is to leverage the strengths of nonlinear SVMs in more complex models where its standard formulation cannot be utilized or fully integrated. For example, discriminative factor models.

We agree that a larger discussion about related work on discriminative feature learning would be an important addition to the paper, however we decided not to do so mainly due to space limitations.

Thanks the first reviewer for noting Peter Sollich’s work. We will definitely cite the paper if our paper is accepted.